# Epigenetic Manipulation Induced Production of Immunosuppressive Chromones and Cytochalasins from the Mangrove Endophytic Fungus *Phomopsis asparagi* DHS-48

**DOI:** 10.3390/md20100616

**Published:** 2022-09-29

**Authors:** Ting Feng, Chengwen Wei, Xiaolin Deng, Dandan Chen, Zhenchang Wen, Jing Xu

**Affiliations:** Collaborative Innovation Center of Ecological Civilization, School of Chemical Engineering and Technology, Hainan University, Haikou 570228, China

**Keywords:** mangrove endophytic fungus, *Phomopsis asparagi*, epigenetic manipulation, chromones, cytochalasins

## Abstract

A mangrove endophytic fungus *Phomopsis asparagi* DHS-48 was found to be particularly productive with regard to the accumulation of substantial new compounds in our previous study. In order to explore its potential to produce more unobserved secondary metabolites, epigenetic manipulation was used on this fungus to activate cryptic or silent genes by using the histone deacetylase (HDAC) inhibitor sodium butyrate and the DNA methyltransferase (DNMT) inhibitor 5-azacytidine (5-Aza). Based on colony growth, dry biomass, HPLC, and ^1^H NMR analyses, the fungal chemical diversity profile was significantly changed compared with the control. Two new compounds, named phaseolorin J (**1**) and phomoparagin D (**5**), along with three known chromones (**2**–**4**) and six known cytochalasins (**6**–**11**), were isolated from the culture treated with sodium butyrate. Their structures, including their absolute configurations, were elucidated using a combination of detailed HRESIMS, NMR, and ECD and ^13^C NMR calculations. The immunosuppressive and cytotoxic activities of all isolated compounds were evaluated. Compounds **1** and **8** moderately inhibited the proliferation of ConA (concanavalin A)-induced T and LPS (lipopolysaccharide)-induced B murine spleen lymphocytes. Compound **5** exhibited significant in vitro cytotoxicity against the tested human cancer cell lines Hela and HepG2, which was comparative to the positive control adriamycin and fluorouracil. Our finding demonstrated that epigenetic manipulation should be an efficient strategy for the induction of new metabolites from mangrove endophytic fungi.

## 1. Introduction

Mangrove endophytic fungi, which adapted to extreme environmental stresses, such as high salinity, high temperature, high humidity, light, and air limitations, are considered to be a reliable source of unique metabolites [1,2,3,4]. Exploring the secondary metabolites with excellent biological activity and pharmacy value from mangrove-derived fungi has become a new hotspot in drug development [5]. Nevertheless, genome sequencing unveils that most mangrove endophytic fungi possess significantly more biosynthetic gene clusters than the number of compounds they produce under conventional culture conditions [6,7,8,9,10]. These facts inspire researchers to develop suitable strategies to stimulate these gene clusters described as ‘silent’, ‘orphan’, and ‘cryptic’ that could, therefore, provide access to an enormous reservoir of structurally novel secondary metabolites to enhance the potential pharmaceutical usage. Several approaches have been successfully used to elicit untapped metabolite profiles, such as OSMAC (One Strain of Many Compounds), which includes media composition, UV irradiation, shaking, incubation temperature, and epigenetic manipulation; and genome mining strategies, which include transcriptional regulator modulation, promoter engineering, and the heterologous expression [11,12,13,14,15]. The methods that use genetic engineering techniques require a relatively sophisticated knowledge of the biology of the producing or surrogate host organisms [16]. In contrast, epigenetic manipulation has been demonstrated to be an effective method for enhancing secondary metabolite expression without altering genes or causing the hereditable manipulation of organisms [17]. There are three main types of small molecule epigenetic regulators known to modulate secondary metabolite expression: DNA methyltransferase (DNMT) inhibitors, 5-azacytidine (5-aza) and *N*-phthalyl-L-tryptophan (RG108); histone deacetylase (HDAC) inhibitors, suberoylanilide hydroxamic acid (SAHA), suberoylbis hydroxamic acid (SBHA), nicotinamide, sodium butyrate, valproic acid, and octanoylhydroxamic acid, and histone acetyltransferase (HAT) inhibitor, and anacardic acid. These inhibitors have been added alone [18,19,20,21,22,23] or in combination [24,25,26] to culture media, successfully inducing or changing the metabolic pathways to enhance the production and/or accumulation of different compounds that are not detected in axenic cultures. For example, the production of cytosporones active against malaria and methicillin-resistant *Staphylococcus aureus* was enhanced, and a previously undescribed cytosporone R was isolated when the histone deacetylase inhibitor (HDAC) sodium butyrate and the DNA methyltransferase (DNMT) inhibitor 5-azacytidine (5-aza) were employed to activate the genes of the marine fungus *Leucostoma persoonii*, an endophyte of mangroves [27]. Baker’s group screened the potential of mangrove-derived endophytic fungi as a source of new antibiotics when cultured in the presence and absence of small molecule epigenetic modulators. Of 1608 extracts from 530 fungal isolates, nearly half (44%) of those fungi producing active extracts only did so following sodium butyrate and 5-aza treatment [28]. These cases might validate that chemical epigenetic manipulation is feasible to efficiently uncover cryptic secondary metabolites from mangrove endophytic fungi. However, the successful examples of epigenetic manipulation applied to mangrove endophytic fungi are limited to confirm the conclusion. 

The coelomycetous genus *Phomopsis* belongs to the family Diaporthaceae and consists of approximately 900 fungal species from a wide range of hosts [29]. The different species belonging to the genus *Phomopsis* are especially known for producing a wide variety of compounds with pharmacological properties, notably cytotoxic [30,31,32], antimicrobial [33,34,35], *β*-site amyloid precursor protein cleaving enzyme 1 (BACE1) inhibitory [36], anti-Tobacco mosaic virus (TMV) [37] and immunosuppressive activities [38]. As part of our research on discovering structurally novel and biologically active natural products from mangrove-derived endophytic fungal strains [39,40,41,42,43,44,45,46,47,48], a strain of *Phomopsis asparagi* DHS-48 isolated from the fresh root of *Rhizophora mangle* attracted our attention for the characterization of a series of immunosuppressive chromones [46] and cytochalasins [38]. In the present study, in order to tap the metabolic potential of this titled fungal strain, epigenetic manipulation was applied to activate its cryptic secondary biosynthetic pathways. The colony growth, dry biomass, ^1^H NMR, and HPLC chromatogram were detected under the cultivation with small molecule epigenetic modifiers, the DNMT inhibitor 5-aza, the HDAC inhibitor sodium butyrate, and a combination of these inhibitors at various concentrations. A follow-up fermentation of an optional modifier (50 µM sodium butyrate) led to the isolation of two new compounds, phaseolorin J (**1**) and phomoparagin D (**5**), along with nine known phaseolorin D (**2**) [49], chaetochromone B (**3**) [50], pleosporalin D (**4**) [51], cytochalasins J, J1, J2, J3, H (**6**–**10**) [31] and phomopchalasin D (**11**) [38]. Herein, we report the epigenetic manipulation of this fungus, and the isolation, structural determination, and bioactivity evaluation of the induced products (Figure 1). A hypothetical biosynthetic pathway for the isolated metabolites is also discussed.

## 2. Results

### 2.1. Epigenetic Manipulation

The epigenetic manipulation of *Phomopsis asparagi* DHS-48 was conducted in both liquid medium and solid medium by using the DNMT inhibitor 5-aza, the HDAC inhibitor sodium butyrate, and the combination of these inhibitors at different concentrations (0, 10, 50, 100 µM). Cultivation without these epigenetic modifiers was used as a control. By comparing the colony growth on PDA (Figure 2a) and dry biomass (calibration graph Figure 2b) in PDA (Figure 2c) and PDB (Figure 2d), we found that the DNMT and HDAC inhibitors produced inconsistent results, and 50 µM sodium butyrate solid fermentation was preferable to induce more remarkable chemical diversity of the secondary metabolites. The HPLC analyses of the EtOAc extracts of *Phomopsis asparagi* DHS-48 cultivated in the presence of different epigenetic agents in all the cases further confirmed our deduction (Appendix A). Consequently, a scaled-up fermentation with 50 µM sodium butyrate was carried out.

The EtOAc extracts of the mycelia and solid rice medium incubated with 50 µM sodium butyrate were subjected to HPLC analyses. By comparing with the blank control (Figure 3 and Appendix A), the production levels of the known metabolites **6**–**8**, **10**, and **11** were considerably enhanced in the sodium-butyrate-inhibited fermentation at the same injection concentration. In addition, certain peaks of **1**–**5** and **9** appear to be present in the chromatograms from the 50 µM HDAC inhibitor that are absent in the control group. Continuously, these differences were also supported by the fact that the ^1^H NMR metabolic profile (Figure 4) of EtOAc extracts showed several additional significant hydrogen resonances between 5.5 and 8.0 ppm, compared with the control group.

### 2.2. Structure Elucidation of the New Compounds

Phaseolorin J (**1**) was isolated as a light yellow amorphous powder. Its molecular formula was determined as C_15_H_16_O_7_ on the basis of HRESIMS data (*m*/*z* 331.0781 [M + Na] ^+^, calcd for C_15_H_16_O_7_ Na 331.0788), which clearly indicated the presence of eight indices of unsaturation. The ^1^H and ^13^C NMR data of **1** (Table 1) and its ^1^H-^1^H COSY and HSQC spectra showed the presence of a series of characteristic signals for a 1,2,3-trisubstituted benzene ring (*δ*_H_ 6.43 (d, *J* = 8.2 Hz), *δ*_C_ 109.3, d, CH-2; *δ*_H_ 7.38 (t, *J* = 8.2 Hz), *δ*_C_ 138.9, d, CH-3; *δ*_H_ 6.52 (d, *J* = 8.2 Hz), *δ*_C_ 108.7, d, CH-4), one olefinic methine of a trisubstituted double bond (*δ*_H_ 5.61 (dq, *J* = 4.8, 1.7 Hz); *δ*_C_ 122.3, d, CH-7), a tertiary methyl (*δ*_H_ 1.86, 3H, d, 1.7; *δ*_C_ 19.2, q, CH_3–_11), an oxygenated methylene (*δ*_H_ 4.12, (d, *J* = 13.2 Hz), *δ*_H_ 4.03 (d, *J* = 13.2 Hz); *δ*_C_ 64.1, t, CH_2_-12), and two oxygenated methines (*δ*_H_ 4.69, 1H, br s, *δ*_C_ 74.5, d, CH-5; 4.55, 1H, d, *J* = 4.8 Hz, *δ*_C_ 67.9, d, CH-8). The magnitude of the ^1^H-^1^H COSY spectrum led to the observation of long-range correlations, including the assignments of vicinal coupling with H-5 and proton H-7 on the *cis*-substituted double bond, as well as homoallylic couplings with H-8 and CH_3_-11. A comparison of the ^1^H and ^13^C NMR data of **1** with those of phaseolorin D (**2**) [52] revealed that both had the same chromone core, except for the presence of one trisubstituted double bond at C-6 (*δ* 140.5) and C-7 (*δ* 122.3) instead of sp^3^ methine (C-6) and sp^3^ methylene (C-7) in **2**. Confirming evidence was obtained from the ^1^H-^1^H COSY correlation from olefinic proton (*δ*_H_ 5.61, H-7) to the oxygenated methine (*δ*_H_ 4.69, H-8) and HMBC correlations from H_3_-11 to C-5, C-6 and C-7 (Figure 5). 

In the NOESY experiment of **1** (Figure 6), the correlations of H_2_-12/H-5 indicated the same spatial orientation. Biogenetically, the configuration of **1** was deduced to be the same as that of **2**, and the calculated ECD spectrum method can be used to predict the absolute configuration of C-8a and C-10a, respectively (Figure 7). Consequently, the absolute configuration of C-5 was assigned to be *S*. However, neither the lack of NOE between H-5/H-8 nor the adjacent coupling constant of *J*_7,8eq_ = 4.8 Hz between H-7 and H-8 supported the relative configuration between H-5 and H-8. To solve this problem, the *δ*_C_ values of two plausible epimers, namely 5*S*,5a*S*,8*S*,8a*R*-**1** and 5*S*,5a*S*,8*R*,8a*R*-**1** (8-*epi*-**1**), were performed after the optimization of the selected conformers at the B3LYP/6-31G(d) level. The results showed that the calculated ^13^C NMR spectrum of the truncated model 5*S*,5a*S*,8*S*,8a*R* -**1** perfectly matched with the experimental one (Figure 8). Therefore, the configuration of **1** was conclusively assigned and given the tentative name phaseolorin J.

Phomoparagin D (**5**) was obtained as a colorless amorphous powder. The molecular formula of **5** was established as C_28_H_37_NO_5_ from its HRESIMS (*m*/*z* 506.2304 [M + K] ^+^, calcd for C_28_H_37_NO_5_K 506.2303. The ^1^H NMR spectrum (Table 2) showed proton signals for a mono substituted phenyl at *δ*_H_ (7.22−7.31, 5H), a tertiary methyl at *δ*_H_ (0.92, 3H, s, H_3_-23), two secondary methyl groups at *δ*_H_ (0.80, 3H, d, *J* = 6.7 Hz, H_3_-11; 1.02, 3H, d, *J* = 6.8 Hz, H_3_-22), an exocyclic methylene group at *δ*_H_ (5.22 and 5.01, 2H, both s, H_2_-12), four oxygenated methine groups at *δ*_H_ (3.82, 1H, d, *J* = 10.8 Hz, H-7; 3.21, 1H, d, *J* = 2.4 Hz, H-19; 2.99, 1H, m, H-20; 3.43, 1H, s, H-21), and two olefinic methine groups at *δ*_H_ (5.72, 1H, d, *J* = 15.5 Hz, 9.5 Hz, H-13; 5.54, 1H, m, H-14). The ^13^C NMR and DEPT spectra (Table 1) of compound **5** displayed 28 carbons, including 3 sp^3^ methyls, 3 sp^3^ methylenes, 9 sp^3^ methines, 2 sp^3^ quaternary carbons, 1 sp^2^ exocyclic methylene, 7 sp^2^ olefinic methines, and 3 sp^2^ quaternary carbons (2 olefinic carbon and 1 amide carbonyl). The carbon profile and characteristic ^1^H NMR signals, as well as the 2D NMR spectra of **5** revealed that it has a similar indole-based cytochalasin skeleton as that of cytochalasin J (**6**), which was first reported in 1981 as deacetylcytochalasin H from the same *Phomopsis* sp. [53]. The main difference between the two compounds is the lack of the typical C_19_-C_20_ double bond (*δ*_H_ 5.76, *δ*_C_ 129.3, d, CH-19; *δ*_H_ 5.85, *δ*_C_ 137.2, d, CH-20) in the macrocycle ring of the latter that was replaced by a 19, 20-epoxide ring (*δ*_H_ 3.21 (d, *J* =2.4 Hz), *δ*_C_ 63.1, CH-19; *δ*_H_ 2.99, m, *δ*_C_ 57.7, CH-20) in **5**. The existence of the epoxide ring was deduced by the analysis of its HRESIMS data, and the molecular weight of **5** was 16 mass units larger than that of **6**. This finding was supported by the ^1^H-^1^H COSY correlations of H-7/H-8/H-13/ H-14/ H-15/ H-16/ H-17/ H-18/ H-19/H-20/H-21, along with the HMBC correlations from H_3_-23 (*δ*_H_ 0.92, 3H, s) to C-17, C-18 and C-19, and H-21 (*δ*_H_ 3.43, 1H, s) to C-8, C-9, C-19 and C-20 (Figure 5). The diagnostic ROESY correlations (Figure 6) positioned H-3, H_3_-11, H-7, H_3_-22, H_3_-23, H-20, and H-21 on the α-face and H-4, H-5, H-8, H-14, H-16, and H-19 on the β-face of **5**, whereas the absolute configuration was assigned by a comparison of the experimental and simulated electronic circular dichroism (ECD) spectra generated by the time-dependent density functional theory (TDDFT) calculations at the B3LYP/6-31+G(d,p) level using the Gaussian 09 program. The experimental ECD spectrum (CH_3_OH) for 3*S*, 4*R*, 5*S*, 7*S*, 8*R*, 9*R*, 16*R*, 16*R*, 19*R*, 20*S*, and 21*R* -**5** matched well with the calculated spectrum (Figure 7), which confirmed the unambiguous assignment of the absolute configuration of **5**, and the trivial name phomoparagin D was assigned. The possible biogenetic pathway of phomoparagin D (**5**) was postulated (Figure 1), which might arise from cytochalasin J (**6**) by a different set of catalyzed reactions.

A plausible biosynthesis of compounds **1**–**11** was proposed, as shown in Figure 1 and Figure 2. More than 4000 chromones have been isolated and structurally elucidated from natural origin until now, and they are biosynthesized by the type III polyketide synthases (PKSs) [54]. Compounds **1** and **2** isolated from *P. asparagi* DHS-48 are assumed to be derived from one acetyl-CoA starter and seven molecules of malonyl-CoA extender units to form an octaketide that undergoes Claisen condensation and cyclization to yield anthraquinone precursors such as emodin, even though it was not isolated in this study. Oxidative cleavage, cyclization via epoxidation, and nucleophilic attack by a hydroxyl group to give the ring-closed dihydroxanthone involved the epimerization of C-10a. The subsequent keto–enol equilibrium and redox would provide compounds **1** and **2**, referring to the reports made by Rönsberg et al. [55]. Previous feeding experiments with sodium ^13^C-labeled acetate by Lösgen et al. [56] in 2007 revealed that a heptaketide precursor is involved in the biosynthesis of **3** and **4**, which are analogues to phomochromenones D-G isolated in our previous study [46], implying some cryptic post-synthesis modification genes were stimulated by the currently adopted epigenetic manipulation for the production of those metabolites previously unobserved or merely increased sufficiently under epigenetic control to be detected. Cytochalasins **5–11** might rationally share a common biosynthetic precursor as we previously described via polyketide synthase (PKS)/nonribosomal peptide synthetase (NRPS) hybrid machinery [38]. The stimulated metabolite **5** was is likely to be also derived from **6** by epoxidation, meanwhile **9** feasibly converted through catalytic dehydration.

### 2.3. Biological Activity of Compounds

The immunosuppressive assay showed that compounds **1** and **8** exhibited moderate-to-weak inhibitory activity against ConA-induced T and LPS-induced B murine splenic lymphocytes in vitro, with the IC_50_ values of 42 and 88 µM and 15 and 110 µM (Table 3), respectively, whereas the other investigated compounds showed no apparent inhibitory effect. Additionally, compound **5** showed significant in vitro cytotoxicity against human cancer cell lines Hela, with an IC_50_ value of 5.8 µM, and showed moderately significant in vitro cytotoxicity against human cancer cell lines HepG2, with an IC_50_ value of 59 µM (Table 4), respectively, which was comparable with the positive controls adriamycin and fluorouracil. These results suggested that the 19,20-epoxide ring in compound 5 is essential for its inhibition of tumor cell proliferation compared with compounds **6**–**11**. 

## 3. Materials and Methods

### 3.1. General Procedures

The specific rotations were obtained on an ATR-W2 HHW5 digital Abbe refractometer (Shanghai Physico-optical Instrument Factory, Shanghai, China). The UV spectra were determined using a Shimadzu UV-2600 PC spectrophotometer (Shimadzu Corporation, Tokyo, Japan), while the ECD spectra were measured on a JASCO J-715 spectra polarimeter (Japan Spectroscopic, Tokyo, Japan). The 1H, 13C, and 2D NMR spectra were recorded on a Bruker AV 400 NMR spectrometer using TMS as an internal standard. High-resolution ESI-MS was performed on an LCMS-IT-TOF instrument (Shimadzu Corporation, Tokyo, Japan) using peak matching. TLC and column chromatography (CC) were carried out over silica gel (200–400 mesh, Qingdao Marine Chemical Inc., Qingdao, China) or Sephadex-LH-20 (18−110 µm, Merck, Darmstadt, Germany), respectively. HPLC analysis was measured on Wasters e2695 (Waters Corporation, Milford, MA, USA) using a C18 column (Waters, 5 μm, 10 × 150 mm). Semi-preparative HPLC was achieved on an Agilent Technologies 1200LC instrument with a C18 column (Agilent Technologies 10 mm × 250 mm). High-speed centrifugation was performed using a TGL-16B Anting centrifuge (Anting Scientific instrument Factory, Shanghai, China). The constant temperature water bath was in HH-2 thermostat water baths (Hervey Biotechnology Corporation, Jinan, China). Liquid fermentation was carried out in an ATL-03202 High-precision CNC shaking machine (Shanghai Kanxin Instrument and Equipment Corporation, Shanghai, China). The purity of the isolated compounds was determined via high-performance liquid chromatography (HPLC), which was performed on an Agilent 1200 instrument and a reverse-phase column (4.6 × 150 mm, 5 μm). The UV wavelength for detection was 210 nm. All the crude extracts and compounds were eluted with a flow rate of 0.8 mL·min^−1^ over a 50 min gradient (solvents: A, H_2_O; B, MeOH), as follows: 0–5min, 25% B; 5–15 min, 25–30% B; 15–30 min, 30–55% B; 30–40 min, 55–75% B; 40–50 min, 70–90% B (Appendix A).

### 3.2. Fungal Material

The endophytic fungi *Phomopsis asparagi* DHS-48 was isolated with a PDA medium from the fresh root of the mangrove plant Rhizophora mangle, collected in October 2015 in Dong Zhai Gang-Mangrove Garden on Hainan Island, China. The strain was isolated under sterile conditions from the inner tissue of the root, following an isolation protocol described previously [57], and the fungi (strain no.DHS-8) was identified using a molecular biological protocol via the DNA amplification and sequencing of the ITS region (GenBank Accession no.MT126606). A voucher strain was deposited at one of the authors’ laboratories (J.X.).

### 3.3. Epigenetic Manipulation and Culture Condition

For the epigenetic manipulation experiments, fungal mycelia and spores were initially inoculated onto Petri dishes containing potato dextrose agar (PDA) at 28 °C for 5 days. Then, a single colony was inoculated into a 100 mL potato dextrose broth (PDB) (in 500 mL Erlenmeyer flasks with continuous shaking for ten days at 28 °C) and the PDA plates (15 mL agar media inverted incubated for five days at 28 °C) were treated with different concentrations (0, 10, 50, and 100 µM) of the DNMT inhibitor 5-aza and the HDAC inhibitor sodium butyrate, or a combination of the two, while the control cultures were treated with vehicle only (filter-sterilized H_2_O). The quantity of biomass is an essential parameter in the determination of a suitable epigenetic modifier or its optimal addition. After filtering the PDB liquid medium, the mycelium precipitate was washed three times with distilled water and lyophilized to constant weight as dry biomass. For the fungi that grow on PDA, the direct measurement of fungal biomass is hampered because the fungi penetrate into and bind themselves tightly to the solid-substrate particles. The indirect method based on the nucleic acid contents was adopted according to Liu’s method [58], with some modifications. The pure mycelium of 0.05, 0.1, 0.15, 0.2, 0.25, and 0.3 g was extracted by adding 25 mL of 5% trichloroacetic acid solution in a water bath at 80 °C for 25 min with constant stirring and then cooled in an ice bath at 8000 r/min, centrifuged at 4 °C for 15 min, and diluted 5 times. The OD value was measured at 260 nm with 1% trichloroacetic acid as the blank control. Finally, dry biomass was quantified based on a standard curve between the nucleic acid content and dry biomass ranging from 0.05 to 0.3 g with y = 4.3543x − 0.0158 (R^2^ = 0.998). All the culture groups were prepared and measured in 3 replicates. The HPLC profiles of the EtOAc extracts of the fungi cultivated in the presence of different epigenetic agents were tested. The cultures were extracted three times with EtOAc (50 mL × 3 for each PDA plate, 250 mL × 3 for each PDB flask). The EtOAc-soluble materials were passed over organic membranes and then subjected to HPLC analysis under conditions mentioned in Section 3.1. 

### 3.4. Extraction Isolation

The fungus was cultivated on PDA by adding 50 µM sodium butyrate at 28 °C for 7 days. Then, a single colony was inoculated in an autoclaved rice solid-substrate medium in Erlenmeyer flasks (130 × 1 L), each containing 100 g of rice, 100 mL of 0.3% of saline water, and 50 µM sodium butyrate and fermented at 28 °C for 28 days. Briefly, 130 flasks of cultures were extracted 3 times with 400 mL of EtOAc, and the filtrate was evaporated under reduced pressure to yield a crude extract of 20 g. The crude extracts were analyzed using HPLC and ^1^H NMR. The EtOAc extracts were chromatographed on silica gel column chromatography (CC) using a step gradient elution process with CH_2_Cl_2_-MeOH (0–100%) to provide nine fractions (Fr. 1–Fr. 9). Fr. 3 was subjected to open silica gel CC using gradient elution with CH_2_Cl_2_-EtOAc (6:1–1:2, *v*/*v*) to yield seven fractions. Fr. 3.1–Fr. 3.7. Fr. 3.4 were purified with semi-preparative reversed-phase HPLC using MeOH-H_2_O (70:30, *v*/*v*) to afford compound **10** (6 mg) (Appendix A) and compound **7** (6 mg) (Appendix A). In addition, Fr. 3.5 was separated via silica gel CC using CH_2_Cl_2_-EtOAc (3:1, *v*/*v*) and purified via semi-preparative reversed-phase HPLC using MeOH-H_2_O (70:30, *v*/*v*) to afford compound **11** (7 mg) (Appendix A). Fr. 4 was separated via open silica gel CC using gradient elution with CH_2_Cl_2_-EtOAc (3:1–1:1, *v*/*v*) to obtain three fractions (Fr. 4.1–Fr. 4.3). Fr. 4.2 was chromatographed on a Sephadex LH-20 CC by eluting with MeOH to yield three fractions (Fr. 4.2.1–Fr. 4.2.3). Fr. 4.2.1 was purified using a silica gel flash column with CH_2_Cl_2_-EtOAc (2:1, *v*/*v*) as the eluent to obtain compound **8** (11.8 mg) (Appendix A) and compound **9** (3 mg) (Appendix A). Fr. 4.2.2 was subjected to Sephadex LH-20 CC using MeOH-CH_2_Cl_2_ (1:1, *v*/*v*) as an eluent to obtain compound **6** (5.9 mg) (Appendix A). Fr. 5 was subjected to open silica gel CC using gradient elution with CH_2_Cl_2_-EtOAc (4:1–1:2, *v*/*v*) to yield five fractions Fr. 5.1–Fr. 5.5. Fr. 5.3 was subsequently subjected to Sephadex LH-20 CC using MeOH-CH_2_Cl_2_ (1:1, *v*/*v*) as an eluent to give six fractions Fr. 5.3.1–Fr. 5.3.6. Fr. 6 was separated through silica gel elution using CH_2_Cl_2_-EtOAc (2:1, *v*/*v*) to obtain six fractions (Fr. 6.1–Fr. 6.6). Fr. 6.3 was purified with Sephadex LH-20 CC using MeOH-CH_2_Cl_2_ (1:1, *v*/*v*) to yield compound **4** (12 mg) (Appendix A). Fr. 6.4 was subjected to reversed-phase HPLC (MeOH-H_2_O 70:30, *v*/*v*) to obtain compound **3** (5 mg) (Appendix A). Fr. 6.5 was subjected to open silica gel CC using gradient elution with CH_2_Cl_2_-EtOAc (4:1–1:2, *v*/*v*) to give five fractions (Fr. 6.5.1–Fr. 6.5.5). Additionally, promising Fr. 6.5.4 was purified with reversed-phase HPLC (MeOH-H_2_O, 70:30 to 100:0, *v*/*v*) to furnish compound **2** (7 mg) (Appendix A) and compound **1** (6 mg) (Appendix A). Fr.7 was subjected to open silica gel CC using gradient elution with CH_2_Cl_2_-EtOAc (1:1, *v*/*v*) to yield six fractions Fr. 7.1–Fr. 7.6. Fr. 7.3 was subjected to Sephadex LH-20 CC using MeOH-CH_2_Cl_2_ (1:1, *v*/*v*) as an eluent to obtain compound **5** (15 mg) (Appendix A). 

Phaseolorin J (**1**): light yellow amorphous powder (MeOH); [α]^20^_D_ +160 (*c* 0.0001, MeOH); UV (MeOH) λ_max_ 214 nm (the absorptions due to aromatic rings); ^1^H and ^13^C NMR data, see Table 1; HRESIMS m/z 331.0781 [M + Na]^+^ (calcd for C_15_H_16_O_7_Na 331.0788).

Phomoparagin D (**5**): colorless amorphous powder (MeOH); [α]^20^_D_ +60 (*c* 0.0001, MeOH); UV (MeOH) λ_max_ 206 nm (the absorptions due to aromatic rings); ^1^H and ^13^C NMR data, see Table 1; HRESIMS *m/z* 506.2304 [M + K]^+^ (calcd for C_28_H_37_NO_5_K 506.2303).

### 3.5. Theory and Calculation Details

Specific Monte Carlo conformational searches were run by employing Spartan’s 14 software using the Merck molecular force field (MMFF). Conformers with a Boltzmann population of over 0.4% were chosen for ECD (Appendix A) and ^13^CNMR (Appendix A) calculations. Then, the conformers were initially optimized at the B3LYP/6-31G(d) level in the gas phase using the PCM polarizable conductor calculation model. The stable conformations obtained at the B3LYP/6-31G(d) level were further used in magnetic shielding constants. The theoretical calculation of ECD was conducted in MeOH using the time-dependent density functional theory (TD-DFT) at the B3LYP/6-31+g (d, p) level for all the conformers of compounds **1** and **5**. The ECD spectra were generated using the program SpecDis 1.6 (University of Würzburg, Würzburg, Germany) and GraphPad Prism 5 (University of California, San Diego, USA) from dipole-length rotational strengths by applying Gaussian band shapes with sigma = 0.3 eV.

### 3.6. Cytotoxicity Assay

HepG2 (liver cancer cell line) and Hela (cervical cancer cell line) were purchased from the Type Culture Collection of the Chinese Academy of Sciences, Shanghai, China. The cells were grown in an RPMI-1640 culture medium. Cytotoxicity against HepG2 cells and HeLa cells was evaluated using the 3-(4,5-dimethylthiazol-2-yl)-2,5-diphenyltetrazolium bromide (MTT) (Sigma-Aldrich, Missouri, St. Louis, MO, USA) method, as described previously [45]. In addition, 5-fluorouracil (5-FU) (Beijing Solarbio Science and Technology Co., Ltd., 99.8%) (Beijing, China) and adriamycin (Shanghai Macklin Biochemical Co., Ltd, 99.8%) (Shanghai, China) were used as positive controls, respectively.

### 3.7. Isolation and Culture of Spleen Lymphocytes

The BALB/c female mice were sacrificed via cervical dislocation, and their spleens were removed aseptically. The splenocytes were washed using RPMI1640 supplemented with penicillin/streptomycin (100 U/mL and 100 μg/mL, respectively) and 10% heat-inactivated FBS, and collected in a centrifuge tube. The erythrocytes were removed for 3 min with an erythrocyte lysis buffer. The cells were plated at a density of 5 × 10^6^ cells/mL or 1 × 10^7^ cells/mL. Cell numbers were performed using a hemocytometer, and cell viability was determined using the trypan-blue dye exclusion technique; cell viability showed more than 95%. The culture media were kept in a humidified atmosphere of 5% CO_2_ at 37 °C.

### 3.8. Cell Activity and Cell Proliferation

In each 96-well cell culture plate, 100 μL of lymphocyte suspension was inoculated with a concentration of 1*10^7^ cells/mL in each well, and the culture was left overnight in a 37 °C, 5% CO_2_ incubator to stabilize the cells. Then, the compounds or positive control (CsA) diluted in a complete medium to different concentrations were added to each well, resulting in the final concentrations of 1, 5, 10, 15, 20, 30, and 40 μM, respectively. The final concentrations of the compounds in the anti-proliferation assay were 20 μM, 35 μM, 50 μM, 70 μM, and 100 μM. After 72 or 48 h incubation in the incubator, the effect of the compounds on the survival rate and anti-proliferation of splenocytes was analyzed using the CCK-8 method.

### 3.9. Statistical Analysis

All the cell data are presented as the mean standard deviation of the means (S.D.), and a one-way analysis of variance (ANOVA) test was used to evaluate the statistical significance of differences between the groups using GraphPad Prism.

## 4. Conclusions

Collectively, the mangrove endophytic fungus *Phomopsis asparagi* DHS-48 was effectively stimulated using an HDAC inhibitor (sodium butyrate) to produce two new compounds, named phaseolorin J (**1**) and phomoparagin D (**5**), along with nine known chromones (**2**–**4**) and cytochalasins (**6**–**11**). All the isolates were evaluated for their immunosuppressive and cytotoxic activities. Among them, compounds **1** and **8** showed moderately inhibitory activity against the proliferation of ConA-induced T and LPS-induced B murine spleen lymphocytes, and compound **5** exerted comparative or better in vitro cytotoxicity against the tested human cancer cell lines than the positive control. Thus, this study demonstrates that epigenetic manipulation appears to have a large potential for enhancing the production and/or accumulation of new chemodiversity from mangrove endophytic fungi.

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
