# Peer review of "Epigenetic Manipulation Induced Production of Immunosuppressive Chromones and Cytochalasins from the Mangrove Endophytic Fungus Phomopsis asparagi DHS-48"

_marinedrugs, 2022, doi:10.3390/md20100616_

Round 1

Reviewer 1 Report

The authors presented the isolation and structures of two new compounds (1 and 5) from the epigenetic manipulated fungus Phomopsis asparagi DHS-48. The manuscript has a merit to be published in Marine Drugs. However, there are some suggestions which would improve the quality of the manuscript.

1.       Line 148; It should not be written as "olefinic proton" as it contains 13C-NMR data. Should be corrected.

2.       Line 149; The authors report the splitting pattern of the H-7 signal as "q, J = 1.7 Hz". I don't understand why H-7 is "q". This is because H-7 exhibits allyl couplings with CH3-11 and H-5, and vicinal coupling with H-8. These couplings are shown in Figure S4. The authors should clearly explain why H-7 becomes "q".

3.       Line 151; The authors report H-5 as "s". However, in Figure S4, H-5 is allyl-coupled with H-7. It is also long-range coupled with CH3-11. The authors should reconfirm this point.

4.       Line 152; The authors report the H-8 signal as "d, J = 4.8 Hz". However, this interpretation is also questionable. Because H-7 is "q, J = 1.7 Hz". Since H-7 and H-8 have a vicinal relationship, their J values should match. Also, in Figure S4, a homoallylic coupling is observed between CH3-11 and H-8. The authors should analyze the 1H-1H COSY spectrum of compound 1 in more detail. In particular, homoallyl coupling depends on the angle between hydrogen atoms, so it can be used for conformational analysis. I believe it is possible to determine the relative configuration of the hydroxyl group at C-8 based on this coupling.

5.       Line 159 (Figure 5); Compounds 1 and 5 are better described on separate sheets. The authors should split compounds 1 and 5. Also, each carbon in compound 1 should be numbered.

6.       Line 160; The NOE description for compound 1 is not consistent with the correlation arrows in Figure 6. Also, due to the low quality of Figure S7, no clear NOE correlation can be observed. In general, the absence of NOEs does not prove that the hydrogens are pointing in opposite directions. The authors should analyze the relative stereochemistry of compound 1 more carefully and in detail.

7.       Line 169 (Figure 6); Compounds 1 and 5 are better described on separate sheets. The authors should split compounds 1 and 5. Also, each carbon in compound 1 should be numbered.

8.       Line 181; compound 1compound 5

9.       Line 196; The authors report that "H-20 is α-face" and "H-19 is β-face" from the ROESY spectrum. However, in compound 5, both H-19 and H-20 are β-face. The authors should explain this point.

10.    Table 2; The authors should summarize the NMR data for compounds 1 and 5 in separate tables. Additionally, compound 1 should be corrected in the table for comparison with the NMR data for known compound 2. Also, compound 5 should be corrected in the table for comparison with the NMR data of known compound 6.

11.    Lines 350 and 353; In the UV spectra of compounds 1 and 5, absorptions due to aromatic rings should also be added. Also IR spectral data for compounds 1 and 5 should be added.

Author Response

Response to Reviewer 1:

Question 1:   Line 148; It should not be written as "olefinic proton" as it contains 13C-NMR data. Should be corrected.

Answer: Thank you! We are very grateful for your reviewing! According to your suggestion, we have corrected ‘ olefinic proton’ to ‘olefinic methine’ in line 48.

Question 2:    Line 149; The authors report the splitting pattern of the H-7 signal as "q, J = 1.7 Hz". I don't understand why H-7 is "q". This is because H-7 exhibits allyl couplings with CH3-11 and H-5, and vicinal coupling with H-8. These couplings are shown in Figure S4. The authors should clearly explain why H-7 becomes "q".

Answer: Thank you for your suggestion! We enlarged the 1HNMR spectrum and found the splitting pattern of H-7 is not acturally symmetrical q, the same as you supposed. So we change “q” to “dq J = 4.8, 1.7 Hz” to be more proper to describe the J2 and J4 coupling of allyl H-7 with H-8 and CH3-11.

Question 3:   Line 151; The authors report H-5 as "s". However, in Figure S4, H-5 is allyl-coupled with H-7. It is also long-range coupled with CH3-11. The authors should reconfirm this point.

Answer: Thank you for your suggestion! We confirmed the correlation of H-5 is allyl-coupled with H-7, and long-range coupled with CH3-11 in the HMBC spectrum. So we enlarged the 1HNMR spectrum and found H-5 is a br s, so we changed the description of H-5 as br s.

 Question 4: Line 152; The authors report the H-8 signal as "d, J = 4.8 Hz". However, this interpretation is also questionable. Because H-7 is "q, J = 1.7 Hz". Since H-7 and H-8 have a vicinal relationship, their J values should match. Also, in Figure S4, a homoallylic coupling is observed between CH3-11 and H-8. The authors should analyze the 1H-1H COSY spectrum of compound 1 in more detail. In particular, homoallyl coupling depends on the angle between hydrogen atoms, so it can be used for conformational analysis. I believe it is possible to determine the relative configuration of the hydroxyl group at C-8 based on this coupling.

Answer: Thank you for your suggestion! Since H-7 and H-8 have a vicinal relationship, we agreed their J values should match and the

splitting pattern of H-7 was changed from “q” to “m”. We also analyze the 1H-1H COSY spectrum of compound 1 in more detail according to your advice as “The magnitude of the 1H-1H COSY spectrum lead to the observation of long range correlations, including the assignments of vicinal coupling with H-5 and proton H-7 on the cis-substituted double bond, as well as homoallylic couplings with H-8 and CH3-11.” added in Line 153-156. The relative configuration of the hydroxyl group at C-8 based on vicinal coupling constant as added in Line 168-171 as “A typical adjacent coupling constant of J7,8eq=4.8 Hz between H-7 and H-8, no homoallylic couplings between H-5 and H-8, located the latter proton in the equatorial position and consequently the 8-OH group axially. Therefore, in agreement with the NOESY data.”

 Question 5:    Line 159 (Figure 5); Compounds 1 and 5 are better described on separate sheets. The authors should split compounds 1 and 5. Also, each carbon in compound 1 should be numbered.

Answer: Thank you for your suggestion! The carbon number of compound 1 was added in Figure 5. Since there are 7 figures already in the manuscript, if we split compounds 1 and 5 in a separate sheet, there would be 9 figures. And in Figure 5, we focus on the HMBC and COSY correlations of these two new compounds; in Figure 6, the NOESY correlations of these two new compounds are described. That format might be more understandable for readers since the limitation of the figure numbers.

 Question 6: Line 160; The NOE description for compound 1 is not consistent with the correlation arrows in Figure 6. Also, due to the low quality of Figure S7, no clear NOE correlation can be observed. In general, the absence of NOEs does not prove that the hydrogens are pointing in opposite directions. The authors should analyze the relative stereochemistry of compound 1 more carefully and in detail.

Answer: Thank you for your suggestion! Considering for the solvability of compound 1, we used to try to measure the NMR spectrum of 1 in DMSO-d6, expected if it could be available for the detection for the NOESY correlation of the -OH groups with stereo adjacent protons. Unfortunately, there are no more -OH signals detected on the HNMR (attached below). So according to your suggestion, we tried to use the J values to testify the relative stereochemistry of H-7 and H-8, H-5 and H-8 as following:“A typical adjacent coupling constant of J7,8eq=4.8 Hz between H-7 and H-8, no homoallylic couplings between H-5 and H-8, located the latter proton in the equatorial position and consequently the 8-OH group axially. Therefore, in agreement with the NOESY data.”

Question 7:   Line 169 (Figure 6); Compounds 1 and 5 are better described on separate sheets. The authors should split compounds 1 and 5. Also, each carbon in compound 1 should be numbered.

 Answer: Thank you for your suggestion! The carbon number of compound 1 was added in Figure 6. Since there are 7 figures already in the manuscript, in Figure 5, we focus on the HMBC and COSY correlations of these two new compounds; in Figure 6, the NOESY correlations of these two new compounds are described. That format might be more understandable for readers since the limitation of the figure numbers.

Question 8:  Line 181; compound 1→compound 5

Answer: Thank you for your significant reminding! We are very grateful to reviewer reviewing the paper so carefully. We are very sorry for our oversight resulting in incorrect writings. According to your suggestion, We have corrected ‘compound 1’ to ‘compound 5’ in line 181. 

Question 9: Line 196; The authors report that "H-20 is α-face" and "H-19 is β-face" from the ROESY spectrum. However, in compound 5, both H-19 and H-20 are β-face. The authors should explain this point.

Answer: Thank you for your suggestion! From the NOESY spectrum, we do not find any correlation of H-20 and H-19, which indicated that H-19 and H-20 are on the opposite position of the epoxide ring. Also it could be deduce from the 3J19,20=2.4 Hz, which was in accordance of the H-C19-C-20-H angle approximately 120°. Thus the reason why we report H-20 is α-face and H-19 is β-face.

Question 10: Table 2; The authors should summarize the NMR data for compounds 1 and 5 in separate tables. Additionally, compound 1 should be corrected in the table for comparison with the NMR data for known compound 2. Also, compound 5 should be corrected in the table for comparison with the NMR data of known compound 6.

 Answer: Thank you for your suggestion! According to your suggestion, we have seperated the NMR data of compounds 1 and 5 and put them in Tables 2 and 3, individually.

Table 2. 1H (400 MHz) and 13C (100 MHz) NMR data of 1 and 2 in CD3OD.

1

2

δC Type

δH (J in Hz)

δC Type

δH (J in Hz)

163.2, C

163.4, C

109.3, CH

6.43, d, 8.2

109.7, CH

6.44, d, 8.2

138.9, CH

7.38, t, 8.2

139.0, CH

7.39, t, 8.2

108.7, CH

6.52, d, 8.2

109.5, CH

6.55, d, 8.2

160.8, C

160.3, C

74.3, CH

4.69, brs

74.6, CH

4.29, m

140.5, C

29.2, CH

2.34, m

122.3, CH

5.61, dq, 4.8,1.7

32.0, CH2

1.57, dt, 14.8, 2.8

2.43, m

67.9, CH

4.55, d, 4.8

68.6, CH

4.44, t, 3.6

86.5, C

85.4, C

197.2, C

196.2, C

108.4, C

108.6, C

74.5, C

76.4, C

19.2, CH3

1.86, d, 1.7

18.1, CH3

1.32, d, 7.8

64.1, CH2

Ha 4.12, d, 13.2

60.5, CH2

Ha 3.83, d, 13.5

Hb 4.03, d, 13.2

Hb 4.28, m

Table 3. 1H (400 MHz) and 13C (100 MHz) NMR data of 5 and 6 in CD3OD.

Position

5

6

δC Type

δH (J in Hz)

δC Type

δH (J in Hz)

1

178.4, C

178.7, C

3

55.3, CH

3.37, dd, 6.1, 3.5

55.1, CH

3.31, m

4

51.9, CH

2.46, dd, 5.1, 3.5

50.8, CH

2.61, m

5

33.8, CH

2.75, m

34.0, CH

2.92, m

6

151.7, C

151.9, C

7

72.7, CH

3.82, d, 10.8

72.5, CH

3.83, d, 10.7

8

45.8, CH

2.89, d, 10.1

50.2, CH

2.88, m

9

56.1, C

54.9, C

10

44.8, CH2

Ha 2.86, dd, 10.3, 6.2

Hb 2.77, m

46.6, CH2

Ha 2.92, m

Hb 2.59, dd, 13.6, 8.2

11

13.8, CH3

0.80, d, 6.7

14, CH3

1.11, d, 6.7

12

113.1, CH2

Ha 5.22, s

113.1, CH2

Ha 5.34, s

Hb 5.01, s

Hb 5.12, s

13

131.2, CH

5.72, dd, 15.5, 9.5

129.4, CH

5.73, dd, 15.5, 9.8

14

135.4, CH

5.54, m

138.7, CH

5.33, m

15

43.7, CH2

Ha 2.05, dd, 12.4, 5.6

Hb 1.65, d, 12.4

45, CH2

Ha 2.02, m

Hb 1.81, m

16

28.7, CH

1.87, m

29.4, CH

1.80, m

17

48.7, CH2

Ha 1.75, dd, 14.8, 3.2

Hb 1.45, dd, 14.8, 3.8

55, CH2

Ha 1.90, dd, 14.4, 3.4

Hb 1.57, dd, 14.4, 3.3

18

73.6, C

75.1, C

19

63.1, CH

3.21, d, 2.4

137.5, CH

5.75, dd, 16.8, 2.2

20

57.7, CH

2.99, m

132.2, CH

6.03, dd, 16.8, 2.6

22

26.2, CH3

1.02, d, 6.8

26.7, CH3

1.04, d, 6.6

23

23.5, CH3

0.92, s

31.8, CH3

1.36, s

1'

138.5, C

137.5, C

2'/6'

131.2, CH

7.22, d, 7.2

131.1, CH

7.15, dd, 8.0, 1.3

3'/5'

129.6, CH

7.31, t, 7.2

129.5, CH

7.32, dd, 8.0, 7.1

4'

128.0, CH

7.25, dt, 7.2, 3.6

127.8, CH

7.24, dd, 7.1, 1.3

Question 11: Lines 350 and 353; In the UV spectra of compounds 1 and 5, absorptions due to aromatic rings should also be added. Also IR spectral data for compounds 1 and 5 should be added.

Answer: Thank you for your suggestion. We have added ‘absorptions due to aromatic rings’ behind the UV spectra of compounds 1 and 5 in lines 350 and 353. We agree that IR spectral data for compounds 1 and 5 should be added. Thank you again for pointing this out and this is an important consideration. However, the infrared spectroscopy test requires a sample mass of several mgs and the sample cannot be recovered. Compounds 1 and 5 were depleted due to testing of biological activity, so we do not have the condition to complete this test.

Reviewer 2 Report

I have evaluated the manuscript of Ting Feng and hereby tender my findings. The paper describes the isolation of eleven compounds from Phomopsis asparagi DHS-48 after culture manipulation by addition of sodium butyrate as an HDAC inhibitor, which effects epigenetic manipulation of the fungus. Of the eleven compounds isolated, two are new.

Overall the manuscript is pretty well written with a reasonable standard of English. I have uploaded an annotated hard-copy that the authors can use to direct corrections. In addition, I have several specific points that I believe need addressing before the manuscript could be considered acceptable:

Line 13, Figure 3: Why are peaks 1 and 5 listed with a *? (I presume that it is because they are the new compounds, in which case please state this in the figure caption.)

Line 149: I do not follow the authors discussion of the coupling constant/multiplicity of H-7. This resonance is listed as a q, J = 1.7 Hz. This multiplicity and coupling value is suitable for H-7 showing allylic coupling to CH3-11. However, H-8 is listed as a d with J = 4.8 Hz, and as drawn the only proton that H-8 can couple to is H-7, so H-7 should be a dq J = 4.8, 1.7 Hz.        

Line 160: There is no text provided that suggests how the relative configuration between H-8a and the other chiral centres was established, and most importantly how H-8a and H-10a were established. H-10a is suggested to be the oppositive configuration compared with other isolates. I think TD-DFT calculation of NMR data for both possible epimers at H-10a and subsequent DP4+ type analysis is needed to confirm the relative configuration of H-8a/H-10a.

Line 182: It’s a minor point but only 26 13C resonances will be detected, not 28 as listed, as C-2’/6’ and C-3’/6’ are identical pairs.

Lines 192 – 205. Please provide some text regarding any similarities for chemical shifts between compound 5 and other related isolates like 6, especially for C-1 to C-9/H-2 to H-6 as this would support the shared relative configurations.

Lines 237 and 240: These are Schemes 1 and 2, not figures 7 and 8. Also, there appears to be a N atom missing from the product of the first step of scheme 2. I also don’t understand why from the alkene to the epoxide is listed as needing dehydration and then cyclization, surely just epoxidation by an epoxidase/mono-oxygenase would do this?

Bioassay data: The authors are claiming significant, and moderate activity and yet most of the IC50’s are in the high tens of µM, which is pretty low level. I would suggest fluorouracil is a bad choice of positive control if its IC50 is 176 µM. Please comment. (Moderate activity would be <10 µM). Also please explain what is meant by the \ symbol in tables 2 and 3 in the captions.

Line 159, Figure 5: Please add some atom labels to structure 1 to aid the reader in following the paragraph text.

Where is the d coupling of H-7?

Line 159, Figure 5: Please add some atom labels to structure 1 to aid the reader in following the paragraph text.

Line 160: There is no text provided that suggests how the relative configuration between H-8a and the other chiral centres was established, and most importantly how H-8a and H-10a were established. H-10a is suggested to be the oppositive configuration compared with other isolates. I think TD-DFT calculation of NMR data for both possible epimers at H-10a and subsequent DP4+ type analysis is needed to confirm the relative configuration of H-8a/H-10a.

Line 182: It’s a minor point but only 26 13C resonances will be detected, not 28 as listed, as C-2’/6’ and C-3’/6’ are identical pairs.

Lines 192 – 205. Please provide some text regarding any similarities for chemical shifts between compound 5 and other related isolates like 6, especially for C-1 to C-9/H-2 to H-6 as this would support the shared relative configurations.

Lines 237 and 240: These are Schemes 1 and 2, not figures 7 and 8. Also, there appears to be a N atom missing from the product of the first step of scheme 2. I also don’t understand why from the alkene to the epoxide is listed as needing dehydration and then cyclization, surely just epoxidation by an epoxidase/mono-oxygenase would do this?

Bioassay data: The authors are claiming significant, and moderate activity and yet most of the IC50’s are in the high tens of µM, which is pretty low level. I would suggest fluorouracil is a bad choice of positive control if its IC50 is 176 µM. Please comment. (Moderate activity would be <10 µM). Also please explain what is meant by the \ symbol in tables 2 and 3 in the captions.

Where is the d coupling of H-7?

Line 159, Figure 5: Please add some atom labels to structure 1 to aid the reader in following the paragraph text.

Line 160: There is no text provided that suggests how the relative configuration between H-8a and the other chiral centres was established, and most importantly how H-8a and H-10a were established. H-10a is suggested to be the oppositive configuration compared with other isolates. I think TD-DFT calculation of NMR data for both possible epimers at H-10a and subsequent DP4+ type analysis is needed to confirm the relative configuration of H-8a/H-10a.

Line 182: It’s a minor point but only 26 13C resonances will be detected, not 28 as listed, as C-2’/6’ and C-3’/6’ are identical pairs.

Lines 192 – 205. Please provide some text regarding any similarities for chemical shifts between compound 5 and other related isolates like 6, especially for C-1 to C-9/H-2 to H-6 as this would support the shared relative configurations.

Lines 237 and 240: These are Schemes 1 and 2, not figures 7 and 8. Also, there appears to be a N atom missing from the product of the first step of scheme 2. I also don’t understand why from the alkene to the epoxide is listed as needing dehydration and then cyclization, surely just epoxidation by an epoxidase/mono-oxygenase would do this?

Bioassay data: The authors are claiming significant, and moderate activity and yet most of the IC50’s are in the high tens of µM, which is pretty low level. I would suggest fluorouracil is a bad choice of positive control if its IC50 is 176 µM. Please comment. (Moderate activity would be <10 µM). Also please explain what is meant by the \ symbol in tables 2 and 3 in the captions.

Where is the d coupling of H-7?

Line 159, Figure 5: Please add some atom labels to structure 1 to aid the reader in following the paragraph text.

Line 160: There is no text provided that suggests how the relative configuration between H-8a and the other chiral centres was established, and most importantly how H-8a and H-10a were established. H-10a is suggested to be the oppositive configuration compared with other isolates. I think TD-DFT calculation of NMR data for both possible epimers at H-10a and subsequent DP4+ type analysis is needed to confirm the relative configuration of H-8a/H-10a.

Line 182: It’s a minor point but only 26 13C resonances will be detected, not 28 as listed, as C-2’/6’ and C-3’/6’ are identical pairs.

Lines 192 – 205. Please provide some text regarding any similarities for chemical shifts between compound 5 and other related isolates like 6, especially for C-1 to C-9/H-2 to H-6 as this would support the shared relative configurations.

Lines 237 and 240: These are Schemes 1 and 2, not figures 7 and 8. Also, there appears to be a N atom missing from the product of the first step of scheme 2. I also don’t understand why from the alkene to the epoxide is listed as needing dehydration and then cyclization, surely just epoxidation by an epoxidase/mono-oxygenase would do this?

Bioassay data: The authors are claiming significant, and moderate activity and yet most of the IC50’s are in the high tens of µM, which is pretty low level. I would suggest fluorouracil is a bad choice of positive control if its IC50 is 176 µM. Please comment. (Moderate activity would be <10 µM). Also please explain what is meant by the \ symbol in tables 2 and 3 in the captions.

Where is the d coupling of H-7?

Line 159, Figure 5: Please add some atom labels to structure 1 to aid the reader in following the paragraph text.

Line 160: There is no text provided that suggests how the relative configuration between H-8a and the other chiral centres was established, and most importantly how H-8a and H-10a were established. H-10a is suggested to be the oppositive configuration compared with other isolates. I think TD-DFT calculation of NMR data for both possible epimers at H-10a and subsequent DP4+ type analysis is needed to confirm the relative configuration of H-8a/H-10a.

Line 182: It’s a minor point but only 26 13C resonances will be detected, not 28 as listed, as C-2’/6’ and C-3’/6’ are identical pairs.

Lines 192 – 205. Please provide some text regarding any similarities for chemical shifts between compound 5 and other related isolates like 6, especially for C-1 to C-9/H-2 to H-6 as this would support the shared relative configurations.

Lines 237 and 240: These are Schemes 1 and 2, not figures 7 and 8. Also, there appears to be a N atom missing from the product of the first step of scheme 2. I also don’t understand why from the alkene to the epoxide is listed as needing dehydration and then cyclization, surely just epoxidation by an epoxidase/mono-oxygenase would do this?

Bioassay data: The authors are claiming significant, and moderate activity and yet most of the IC50’s are in the high tens of µM, which is pretty low level. I would suggest fluorouracil is a bad choice of positive control if its IC50 is 176 µM. Please comment. (Moderate activity would be <10 µM). Also please explain what is meant by the \ symbol in tables 2 and 3 in the captions.

Where is the d coupling of H-7?

Line 159, Figure 5: Please add some atom labels to structure 1 to aid the reader in following the paragraph text.

Line 160: There is no text provided that suggests how the relative configuration between H-8a and the other chiral centres was established, and most importantly how H-8a and H-10a were established. H-10a is suggested to be the oppositive configuration compared with other isolates. I think TD-DFT calculation of NMR data for both possible epimers at H-10a and subsequent DP4+ type analysis is needed to confirm the relative configuration of H-8a/H-10a.

Line 182: It’s a minor point but only 26 13C resonances will be detected, not 28 as listed, as C-2’/6’ and C-3’/6’ are identical pairs.

Lines 192 – 205. Please provide some text regarding any similarities for chemical shifts between compound 5 and other related isolates like 6, especially for C-1 to C-9/H-2 to H-6 as this would support the shared relative configurations.

Lines 237 and 240: These are Schemes 1 and 2, not figures 7 and 8. Also, there appears to be a N atom missing from the product of the first step of scheme 2. I also don’t understand why from the alkene to the epoxide is listed as needing dehydration and then cyclization, surely just epoxidation by an epoxidase/mono-oxygenase would do this?

Bioassay data: The authors are claiming significant, and moderate activity and yet most of the IC50’s are in the high tens of µM, which is pretty low level. I would suggest fluorouracil is a bad choice of positive control if its IC50 is 176 µM. Please comment. (Moderate activity would be <10 µM). Also please explain what is meant by the \ symbol in tables 2 and 3 in the captions.

Where is the d coupling of H-7?

Line 159, Figure 5: Please add some atom labels to structure 1 to aid the reader in following the paragraph text.

Line 160: There is no text provided that suggests how the relative configuration between H-8a and the other chiral centres was established, and most importantly how H-8a and H-10a were established. H-10a is suggested to be the oppositive configuration compared with other isolates. I think TD-DFT calculation of NMR data for both possible epimers at H-10a and subsequent DP4+ type analysis is needed to confirm the relative configuration of H-8a/H-10a.

Line 182: It’s a minor point but only 26 13C resonances will be detected, not 28 as listed, as C-2’/6’ and C-3’/6’ are identical pairs.

Lines 192 – 205. Please provide some text regarding any similarities for chemical shifts between compound 5 and other related isolates like 6, especially for C-1 to C-9/H-2 to H-6 as this would support the shared relative configurations.

Lines 237 and 240: These are Schemes 1 and 2, not figures 7 and 8. Also, there appears to be a N atom missing from the product of the first step of scheme 2. I also don’t understand why from the alkene to the epoxide is listed as needing dehydration and then cyclization, surely just epoxidation by an epoxidase/mono-oxygenase would do this?

Bioassay data: The authors are claiming significant, and moderate activity and yet most of the IC50’s are in the high tens of µM, which is pretty low level. I would suggest fluorouracil is a bad choice of positive control if its IC50 is 176 µM. Please comment. (Moderate activity would be <10 µM). Also please explain what is meant by the \ symbol in tables 2 and 3 in the captions.

Where is the d coupling of H-7?

Line 159, Figure 5: Please add some atom labels to structure 1 to aid the reader in following the paragraph text.

Line 160: There is no text provided that suggests how the relative configuration between H-8a and the other chiral centres was established, and most importantly how H-8a and H-10a were established. H-10a is suggested to be the oppositive configuration compared with other isolates. I think TD-DFT calculation of NMR data for both possible epimers at H-10a and subsequent DP4+ type analysis is needed to confirm the relative configuration of H-8a/H-10a.

Line 182: It’s a minor point but only 26 13C resonances will be detected, not 28 as listed, as C-2’/6’ and C-3’/6’ are identical pairs.

Lines 192 – 205. Please provide some text regarding any similarities for chemical shifts between compound 5 and other related isolates like 6, especially for C-1 to C-9/H-2 to H-6 as this would support the shared relative configurations.

Lines 237 and 240: These are Schemes 1 and 2, not figures 7 and 8. Also, there appears to be a N atom missing from the product of the first step of scheme 2. I also don’t understand why from the alkene to the epoxide is listed as needing dehydration and then cyclization, surely just epoxidation by an epoxidase/mono-oxygenase would do this?

Bioassay data: The authors are claiming significant, and moderate activity and yet most of the IC50’s are in the high tens of µM, which is pretty low level. I would suggest fluorouracil is a bad choice of positive control if its IC50 is 176 µM. Please comment. (Moderate activity would be <10 µM). Also please explain what is meant by the \ symbol in tables 2 and 3 in the captions.

Where is the d coupling of H-7?

Line 159, Figure 5: Please add some atom labels to structure 1 to aid the reader in following the paragraph text.

Line 159, Figure 5: Please add some atom labels to structure 1 to aid the reader in following the paragraph text.

Author Response

Response to Reviewer 2:

Reviewer’s Comments: I have evaluated the manuscript of Ting Feng and hereby tender my findings. The paper describes the isolation of eleven compounds from Phomopsis asparagi DHS-48 after culture manipulation by addition of sodium butyrate as an HDAC inhibitor, which effects epigenetic manipulation of the fungus. Of the eleven compounds isolated, two are new.

Overall the manuscript is pretty well written with a reasonable standard of English. I have uploaded an annotated hard-copy that the authors can use to direct corrections. In addition, I have several specific points that I believe need addressing before the manuscript could be considered acceptable:

Answer: Thank you very much for your positive comments! We would try our best to address every points raised by the respected reviewer!

Question 1: Line 13, Figure 3: Why are peaks 1 and 5 listed with a *? (I presume that it is because they are the new compounds, in which case please state this in the figure caption.)

Answer: Thank you very much and we have revised according to your suggestion. The peaks 1 and 5 listed with“*”because they are the new compounds. We want to use this way to specifically label the new stimualated new metabolites. So in the figure 3 caption, we have added the statement that “★Compounds 1 and 5 in (a) represent the new compounds stimulated by epigenetic manipulation.”.

Question 2: Line 149: I do not follow the authors discussion of the coupling constant/multiplicity of H-7. This resonance is listed as a q, J = 1.7 Hz. This multiplicity and coupling value is suitable for H-7 showing allylic coupling to CH3-11. However, H-8 is listed as a d with J = 4.8 Hz, and as drawn the only proton that H-8 can couple to is H-7, so H-7 should be a dq J = 4.8, 1.7 Hz.     

Answer: Thank you very much for your suggestions! We enlarged the 1HNMR spectrum and found the splitting pattern of H-7 is not an exactly symmetrical q, the same as you supposed. So we changed “q” to “dq J = 4.8, 1.7 Hz” to be more proper to describe the coupling of J2 and J4 coupling of allyl H-7 with H-8 and CH3-11.

Response to PDF comments:

Question 1: In lines 7 and 8, why list 3 affiliations ?

Answer: Thank you for your comment! These are not 3 different individual affiliations. As for ‘Collaborative Innovation Center of Ecological Civilization’ is attached to ‘School of Chemical Engineering and Technology’ and the ‘School of Chemical Engineering and Technology’ is attached to ‘Hainan University’. Hopefully our explanation satisfied the query.

Question 2: In line 40-line 42,” Nevertheless, genome sequencing unveils that most mangrove endophytic fungi possess significantly more biosynthetic gene clusters than the number of compounds they produced under conventional cultures conditions. These facts inspire the researchers to develop suitable strategies to stimulate these gene clusters described as ‘silent’, ‘orphan’ and ‘cryptic’ could therefore provide access to an enormous reservoir of structurally novel secondary metabolites to enhance the potential pharmaceutical usage.”

Answer: Thank you for your revision. This paragraph has revised according to your suggestion as “Nevertheless, genome sequencing unveils that most mangrove endophytic fungi possess significantly more biosynthetic gene clusters than the number of compounds they produce under conventional cultures conditions [6-10].These facts inspire researchers to develop suitable strategies to stimulate these gene clusters described as ‘silent’, ‘orphan’ and ‘cryptic’ that could therefore provide access to an enormous reservoir of structurally novel secondary metabolites to enhance the potential pharmaceutical usage.”.

Question 3: Line 53-74 grammar mistakes

Answer: Thank you very much for your revision! This paragraph was revised according to your suggestion as ” There are three main types of small molecule epigenetic regulators known to modulate secondary metabolite expression, DNA methyltransferase (DNMT) inhibitors, 5-azacytdine (5-aza) and N-phthalyl-L-tryptophan (RG108); histone deacetylase (HDAC) inhibitors, suberoylanilide hydroxamic acid (SAHA), suberoylbis hydroxamic acid (SBHA), nicotinamide, sodium butyrate, valproic acid, and octanoylhydroxamic acid, and histone acetyltransferase (HAT) inhibitor, anacardic acid. These inhibitors have been added alone [18-23] or in combination [24-26] to culture media, successfully in inducing or changing the metabolic pathways to an enhanced production and/or to an accumulation of different compounds that are not detected in axenic cultures. For example, the production of cytosporones active against malaria and methicillin-resistant Staphylococcus aureus was enhanced and a previously undescribed cytosporone R was isolated when histone deacetylase inhibitor (HDAC) sodium butyrate, and a DNA methyltransferase (DNMT) inhibitor, 5-azacytidine (5-aza) were employed to activate the genes of the marine fungus Leucostoma persoonii, an endophyte of mangroves [27]. Baker’s group screened the potential of mangrove-derived endophytic fungi as a source of new antibiotics when cultured in the presence and absence of small molecule epigenetic modulators. Of 1608 extracts from 530 fungal isolates, nearly half (44%) of those fungi producing active extracts only did so following sodium butyrate and 5-aza treatment [28]. These cases might validate that chemical epigenetic manipulation is feasible to efficiently uncover cryptic secondary metabolites from mangrove endophytic fungi. However, the successful examples of epigenetic manipulation applied to mangrove endophytic fungi are limited to confirm the conclusion. ”

Question 4: Line 77 grammar mistake

Answer: Thank you very much for your revision! “is” was changed to “are”.

Question 5: Line 88-96 grammar mistakes

Answer: Thank you very much for your revision! This paragraph was revised according to your suggestion as “The colony growth, dry biomass, 1H NMR, and HPLC chromatogram were detected under the cultivation with small molecule epigenetic modifiers, DNMT inhibitor 5-aza, HDAC inhibitor sodium butyrate, and a combination of these inhibitors at various concentrations. A follow-up fermentation of the optional modifier (50 µM sodium butyrate) led to the isolation of two new compounds, phaseolorin J (1) and phomoparagin D (5), along with nine known phaseolorin D (2) [49], chaetochromone B (3) [50], pleosporalin D (4) [51], cytochalasins J, J1, J2, J3, H (6-10) [52] and phomopchalasin D (11) [38]. Herein, we report the epigenetic manipulation on this fungus, and the isolation, structural determination, and bioactivity evaluation of the induced products (Figure 1). A hypothetical biosynthetic pathway for the isolated metabolites is also discussed.”

Question 6: Line 101-110 grammar mistakes

Answer: Thank you very much for your revision! This paragraph was revised according to your suggestion as “The epigenetic manipulation of Phomopsis asparagi DHS-48 was conducted in both liquid medium and solid medium by using DNMT inhibitor 5-aza, HDAC inhibitor sodium butyrate and the combination of these inhibitors at different concentrations (0,10,50,100 µM). Cultivation without epigenetic modifiers was used as a control. By comparing colony growth on PDA (Figure 2a), dry biomass (Calibration graph Figure 2b) in PDA (Figure 2c) and PDB (Figure 2d), we found the DNMT and HDAC inhibitors produced inconsistent results and 50 µM sodium butyrate solid fermentation was preferable to induce more remarkable chemical diversity of the secondary metabolites. The HPLC analyses of the EtOAc extracts of Phomopsis asparagi DHS-48 cultivated in the presence of different epigenetic agents in all cases further confirmed our deduction (Figure S44). ”

Question 7: The coordinate and caption mistakes

Answer: Thank you very much for your revision! Figure 2b was revised as following:

The “5-Aza” in caption was revised as 5-aza according to your suggestion.

Question 8: Line 128-130 grammar mistakes

Answer: Thank you very much for your revision! This sentence was revised according to your suggestion as “Continuously, these differences were also supported by the fact that 1H NMR metabolic profile (Figure 4) of EtOAc extracts showed several additional significant hydrogen regonances between 5.5 and 8.0 ppm compared with the control group.”

Question 9: In Figure 3; Why are peaks 1 and 5 listed with a *?

Answer: Thank you for your comment!  The peaks 1 and 5 listed with“*”because they are the new compounds. We want to use this way to specifically label the new stimualated new metabolites. So in the figure 3 caption, we have added the statement that “★Compounds 1 and 5 in (a) represent the new compounds stimulated by epigenetic manipulation.”.

Question 10: The mistyping in caption of Figure 4

Answer: Thank you very much for your revision! The caption as revised as “Figure 4. 1H NMR spectra of EtOAc extracts of Phomopsis asparagi DHS-48 measured in CD3OD at 400 MHz, chemical shifts (δ) presented in ppm.” according to your suggestion.

Question 11: In line 149; A tertiary methyl (δH 1.86, 3H, s; dC 19.2, q, CH3-11) → a tertiary methyl (δH 1.86, 3H, d=1.7 Hz; dC 19.2, q, CH3-11)

Answer: Thank you for your suggestion! The coupling constant in line 149 was change to “a tertiary methyl (δH 1.86, 3H, d=1.7 Hz; dC 19.2, q, CH3-11)” according to your suggestion.

Question 12: In lines 149 and 152; The authors should clearly explain coupling analysis for H-7/H-8.

Answer: Thank you very much for your suggestions! According to your suggestion, we revised the coupling constant of H-7 from “q” to “dq J = 4.8, 1.7 Hz”. We also add some detailed analysis of the COSY spectrum as following: “The magnitude of the 1H-1H COSY spectrum lead to the observation of long range correlations, including the assignments of vicinal coupling with H-5 and proton H-7 on the cis-substituted double bond, as well as homoallylic couplings with H-8 and CH3-11.” We also use the coupling constant of H7/H-8 to analyze the stereochemistry of C-8 as “A typical adjacent coupling constant of J7,8eq=4.8 Hz between H-7 and H-8, no homoallylic couplings between H-5 and H-8, located the latter proton in the equatorial position and consequently the 8-OH group axially. Therefore, in agreement with the NOESY data.” In Line 168-171.

Question 13: In Figure 5; Each carbon in compound 1 should be numbered.

Answer: Thank you for your suggestion! The carbon number of compound 1 was added in Figure 6.

Question 13: In lines 162 and 165 typing mistakes

Answer: Thank you for your suggestion! The sentence in line 162 and 165 was revised to “The absolute configuration of 1 was theoretically deduced to be the same as that of 2 from the biogenetic consideration and confirmed with the aid of the calculated ECD spectrum method, where the calculated ECD spectrum of the truncated model 5S,5aS,8S,8aR-1 perfectly matched with the experimental one (Fig. 7).”

Question 14: In line 182; the 13C NMR spectrum can only display 26 carbon signals?

Answer: Thank you for your suggestion! According to your suggestion, we have double-checked the 13C NMR (Figure S10) and DEPT spectrum (Figure S11), compound 5 indeed displayed 28 carbons. A total of 26 carbon signals appear in the 13C NMR spectrum (Figure S10), δ 178.4 (C-1), 151.7 (C-6), 138.5 (C-1'), 135.4 (CH-14), 131.2 (CH-13), 131.1 (CH-2', 6'), 129.6 (CH-3', 5), 128.0 (CH-4'), 113.1 (CH2-12), 75.5 (CH-21), 73.6 (C-18), 72.7 (CH-7), 63.1 (CH-19), 57.7 (CH-20), 56.1 (C-9), 55.3 (CH-3), 51.9 (CH-4), 48.8 (CH2-17), 45.8 (CH-8), 43.7 (CH2-15), 33.9 (CH-5), 28.7 (CH3-23), 26.2 (CH-16), 23.5 (CH3-22), 13.8 (CH3-14), where δ 131.1 (CH-2', 6') and δ 129.6 (CH-3', 5) represent two signals respectively. Because of the overlap between signal δ 44.8 (CH2-10) and solvent peak, signal δ 44.8 (CH2-10) is not shown in the 13C NMR spectrum (Figure S10). In the DEPT spectrum (Figure S11) the signal δ 44.8 (CH2-10) can be shown, and this is also consistent with an inverted peak (negative signal) for methylene. In conclusion, the 13C NMR (Figure S10) and DEPT (Figure S11) spectrum of compound 5 displayed 28 carbons in total, including three sp3 methyls, three sp3 methylenes, nine sp3 methines, two sp3 quaternary carbon, one sp2 exocyclic methylene, seven sp2 olefinic methines, and three sp2 quaternary carbons (two olefinic carbon and one amide carbonyl). Hopefully our explanation satisfied the question. The grammar mistakes in Line 173-180 were revised according to your revision.

Question 15: In line 190; The authors should give the J values.

Answer: Thank you for pointing this out! We are very sorry for our negligence
of those J values. According to your suggestion, we have added the above J values in line 190 as “[dH 3.21 (d, J =2.4 Hz), dC 63.1, CH-19; dH 2.99, m, dC 57.7, CH-20]”.

Question 16: In lines 192-197; How similar are the chemical shifts to compound 6? Good evidence of same configurations should be given.

Answer: Thank you for your suggestion! In order to give more clearly illustration of chemical shifts, we separated the NMR data of compounds 1 and 5 in Table 2. The comparation of the NMR data of compounds 1 and 2 could be found in Table 2, compounds 5 and 6 could be found in Table 3 in the manuscript at the moment.

Question 17: In line 218, the fungal name should be Italic.

Answer: Thank you for your suggestion! The sentence has been revised as “Compounds 1 and 2 isolated from P. asparagi DHS-48…”

Question 18: In line 237, Figure 7 description

Answer: Thank you for your suggestion! According to your suggestion, we have corrected ‘Figure 7’ to ‘Scheme 1’ in line 237.

Question 19: Proposed Biosynthetic Pathway for compounds 5 and 9 from 6.

Answer: Thank you for your suggestion! We have corrected ‘Figure 8’ to ‘Scheme 2’ in line 240. The biosynthetic pathway for compounds 5 and 9 from 6 were revised according to your suggestion, it is more smart!

Question 20: In line 246 and 247, IC50 values of 59 µM not very significant.

Answer: Thank you for your suggestion! Our definition of the bioactivity significant was compared the value with positive controls Adriamycin (do not displayed any activity) and Fluorouracil (IC50 =176±28.8µM). In comparation with the positive controls, we conclude the cytotoxicity of compound 5 (IC50 = 59 µM) is significant.

Question 21: In Table 2 and 3, what is meant by “/”? Inactive ? Not tested?

Answer: Thank you! We use “-” to take instead of “/”, which means the tested compounds are inactive at the concentration of 200 µM. And we also added the statement below the Tables as “‘-’ stands for no inhibitory effect at 200 µM.” according to your suggestion.

Question 22: In line 257, 258, 267, 290, 292, 293, 314, 318, 355, 359, 373, 388, 406 and 407 grammar mistakes

Answer: Thank you very much! According to your suggestion, we corrected the above grammatical errors, incorrect writings and modified terminologies throughout the text. We are very grateful to reviewer reviewing the paper so carefully. Your efforts on correcting the spellings and grammar errors, specialized vocabularies and polishing the whole manuscript are very much appreciated!

Round 2

Reviewer 2 Report

The authors have made a number of corrections which I appreciate, but I am still not fully convinced by their stereochemical arguments for determining the relative configurations of compound 1. The absence of a homo-allylic coupling between H-5 and H-8 does not tell us anything regarding the relative configuration of the compound, nor does the 4.8 Hz between H-7 and H-8; if the configuration at C-8 was swapped as shown, as C-7 is sp2 hybridized it would likely have a similar coupling constant.

More importantly, the authors have NOT addressed my concerns that they have provided NO evidence for the relative configurations of C-8a and C-10a, which is a major variation from other members of the compound series. Either the authors need to provide more reasoning for this, or they should perform calculations of the NMR data and confirm that the configurations shown are consistent with the data.

Other minor corrections needing attending to are:

1. Line 130, it should be resonances, not regonances

2. Lines 154 to 157: I do not believe it is the magnitude of the COSY spectrum that is important, it is the magnitude of the 1H-1H coupling that is important. Also, lead is misspelt, it should be led.
